# Long-Term Trends of Asthma Mortality in China from 2000 to 2019: A Joinpoint Regression and Age-Period-Cohort Analysis

**DOI:** 10.3390/healthcare10020346

**Published:** 2022-02-11

**Authors:** Guimin Huang, Junting Liu, Tao Li, Dongqing Hou, Wenqian Liu, Yixuan Xie, Tong Zhang, Yijing Cheng

**Affiliations:** Child Health Big Data Research Center, Capital Institute of Pediatrics, Beijing 100020, China; guiminhuang@163.com (G.H.); winnerljt@163.com (J.L.); socott@126.com (T.L.); dqhou@sina.com (D.H.); 13811325178@163.com (W.L.); xie1xuan@hotmail.com (Y.X.); zt@chinawch.org.cn (T.Z.)

**Keywords:** asthma, mortality, age-period-cohort (APC) analysis, joinpoint regression analysis

## Abstract

Background: Trends of asthma mortality vary widely all over the world, while the trends in China over the past 15 years are unknown. The aim of this study was to assess the trends of asthma mortality in China. Methods: Asthma deaths and demographic characteristics were collected from National Death Cause Datasets of Disease Surveillance System between 2004 and 2019. The data were analyzed with joinpoint regression analysis and age-period-cohort (APC) analysis for the mortality rate due to asthma in China. Results: Asthma mortality declined from 2.4 (95% confidence interval (CI): 2.3–2.5) per 100,000 in 2004 to 1.6 (95% CI: 1.5–1.7) per 100,000 in 2019. Age-adjusted asthma mortality rates decreased for men and women in urban and rural areas from 2004 through 2019. The decreasing trend of the mortality rate has slowed down substantially during 2007 and 2009. After that, the decreasing trend has stabilized. The asthma mortality rates generally have a positive relationship with the age of the population when controlling for period and cohort. The period trend decreased and then increased when controlling for age and cohort. Conclusions: We should pay more attention to asthma management plans or treatment for aging people who are facing higher risk of asthma death.

## 1. Introduction

Asthma is a common chronic airway disease worldwide [1]. In China, asthma is a major public health challenge, with a national prevalence of asthma of 4.2%, representing 45.7 million Chinese adults [2]. A large mental, emotional, physical, social, and economic burden is placed on asthma patients and their relatives [1]. Trends in asthma mortality provide a useful barometer of the impact of changes in asthma management and associated burdens. The locally weighted scatter plot smoother (LOESS) estimate of the global asthma mortality rate in the 5–34-year age group was 0.44 deaths per 100,000 people in 1993 and 0.19 deaths per 100,000 people in 2006, while there was no appreciable change from 2006 through 2012, which was 0.19 deaths per 100,000 people [3]. However, the global asthma mortality rate estimation did not include China. In a previous study, China had a low prevalence of asthma but a relatively high case fatality rate [4]. In recent decades, China has developed into an industrialized country with large populations in megacities, and the accompanying challenges of air pollution from a traditional agricultural society, as well as the aging of the population, could affect the mortality of asthma [5]. Timely and reliable information on cause-specific mortality is central for informing the development, implementation, and evaluation of health policy [6]. However, trends in asthma mortality in China over the past 15 years are unknown. We aim to examine the trend of asthma mortality for all ages by gender and by district from 2004 to 2019 and how age, period, and birth cohorts affect the asthma mortality rate.

## 2. Materials and Methods

### 2.1. Data Sources

The asthma mortality and related data for this study were extracted from the National Death Cause Datasets of Disease Surveillance System written by Chinese Center for Disease Control and Prevention. The system is considered to be representative nationally as it is a population-based death registration system containing 300 million people across 31 provinces using an iterative method involving multistage stratification [7]. The system has increased the surveillance population from 6% to 24% of the Chinese population since 2013, while the number of surveillance points has increased from 161 to 605.

### 2.2. Statistics

Joinpoint regression analysis and age-period-cohort (APC) analysis were performed in this study. The age-adjusted mortality rate was calculated for each age group by sex, from 0, 1–4, 5–9, etc., to 85–89. These data were used to perform the joinpoint regression analysis. The annual data for this study were then aggregated in three 5-year periods: 2004–2008, 2009–2013, and 2014–2018. The 18 age groups of the population were arranged from 0–4, 5–9, etc., until 85–89. The 20 birth cohorts were calculated from period minus age, starting from 1919–1923, 1924–1928, etc., to 2014–2018. The deaths, total population, and mortality due to asthma for each age, period, and birth cohort group by sex are aggregated in (Appendix A). These data were used for the age-period-cohort (APC) analysis.

#### 2.2.1. Joinpoint Regression Analysis

The continuous changes in asthma mortality in China by gender and by district were described by joinpoint regression models. In the models, logarithmic transformation of the mortality rates was carried out, and binomial approximation was used to calculate the standard errors [8]. The average annual percent changes (AAPC) and corresponding 95% confidence interval (CI) were estimated by joinpoint regression analysis to measure the magnitude of the trend in the asthma mortality rate by gender (male/female) and by district (urban/rural areas) [9]. Each *p* value is created using Monte Carlo methods for each submodel, and the overall asymptotic significance level is maintained through a Bonferroni correction. AAPC was calculated as a geometrically weighted average of different annual percent change values from this analysis. Joinpoint regression analysis was performed using Joinpoint Regression Program 4.9.0.0 developed by the Surveillance Research Program of the US National Cancer Institute.

#### 2.2.2. Age-Period-Cohort (APC) Analysis with the Intrinsic Estimator Method

APC analysis was conducted using three time-varying elements, age, period, and cohort, to better understand the trend of the asthma mortality rate in China. Age (A) effects are the age differences at specific times of the interesting observations; period (P) effects are the differences in the time period of targeted observations; and the effects of differences in the year of birth are named cohort (C) effects [10]. A systematic study of such data is termed age-period-cohort (APC) analysis. APC analysis has the unique ability to determine the entire complex of social, historical, and environmental factors that simultaneously affect the entire population of individuals. It has thus been widely used to address questions of continuing importance to studies of social change, the development of various diseases, aging, and population processes and dynamics. This study used APC analysis with an intrinsic estimator (IE) to learn how the A, P, and C effects relate to the mortality rate of asthma for the population ages from 0 to 85 from 2004 to 2019 in China. The APC analysis was performed in Stata 16.0 software (StataCorp, College Station, TX, USA) using the APC-IE package.

We assumed the number of deaths caused by asthma in an age group at a period time to be a Poisson distribution. An offset of log-transformed population variables was used to account for population growth of each age group. The age-period-cohort model has predictor of the form
µ_ik_ = α_i_ + β_j_ + γ_k_ + δ,(1)
where i, j, and k are indices for age, period, and cohort, which are linked so that j = i + k − 1.

The age group indicated varied risks associated with different age groups. Age variations are linked to biological and social processes of aging. In the general population, people at older age may have a higher risk of mortality because of the natural process of aging and are more vulnerable to complications [11]. The period group indicated that the mortality rate in all age groups changed simultaneously over time. Variations in the period may be caused by external factors that affect people from all age groups equally during a specific time period, such as environmental, social, and economic factors. The birth cohort group indicated characteristics specific to the cohort. These variations stem from the unique exposure the cohort experienced as they move across in time, such as occupation of family members, diet habits, etc. The APC-IE method presented the estimated coefficients for the age, period, and cohort effects in the models. The mortality relative risk (RR) of a particular age, period, and cohort relative to each average level was presented using the exponential value of the coefficients (exp(coef.) = ecoef.).

The best-fitting model was defined as the model that minimized the Akaike information criterion (AIC). Models with lower residual deviance had better goodness of fit indicated by lower AIC (Akaike information criterion, lower values indicate better goodness of fit). The significance level was set up at 0.05 (2-sided tests).

## 3. Results

### 3.1. Descriptive Analysis of Asthma Mortality

Overall, the asthma mortality rate dramatically decreased from 2005 to 2006 and steadily decreased thereafter. Up until 2012, the mortality rate decreased by approximately 53% from 2005 (from 2.60 to 1.23 deaths per 100,000 people per year) for males and reduced by approximately 57% (from 2.37 to 1.01 deaths per 100,000 people per year) for females. From 2012 to 2013, the asthma mortality rate increased by approximately 38% for males (from 1.23 to 1.70 deaths per 100,000 people per year), increased by approximately 32% for females (from 1.01 to 1.33 deaths per 100,000 people per year), and plateaued after 2014. The mortality rate in males was generally higher than that in females for the population studied from 2004 to 2019 (Table 1). People in urban areas had a higher mortality rate than people in rural areas.

### 3.2. Joinpoint Regression Analysis

For both males and females from 2004 to 2019, the age-adjusted mortality rate of asthma decreased rapidly from 2004 to 2008 (male annual percent change = −17.1; 95% CI = −25.7, −7.5; female annual percent change = −18.9, 95% CI = −27.9, −8.8), and there was a significant change in 2008. There was a stable and mild decrease from 2008 to 2019 (Figure 1, Table 2). The AAPC for males and females from 2004 to 2019 was −6.0 (95% CI = −8.7, −3.3) and −7.1 (95% CI = −9.9, −4.1), respectively. 

In urban areas, the age-adjusted mortality rate of asthma decreased rapidly from 2004 to 2009 (annual percent change = −12.3; 95% CI = −17.9, −6.2), and there was a significant change in 2009. From 2009 to 2019, the annual percent change = −3.4, 95% CI = −5.2, −1.6 (Figure 2, Table 2). In addition, the AAPC was −6.5, 95% CI = −8.6, −4.3. In rural areas, a significant change occurred in 2007. The annual percent change from 2004 to 2007 was −26.1, a much sharper decrease than urban areas, and then a steady moderate decrease from 2007 to 2019. The AAPC from 2004 to 2019 was −6.7, 95% CI = −11.2, −1.9.

### 3.3. Age-Period-Cohort (APC) Analysis with the Intrinsic Estimator Method

Figure 3 presents the age, period, and cohort effect of asthma mortality for males/females and urban/rural areas. After controlling for period and cohort effects (Figure 3a), the age effect on asthma mortality showed that the relative risk of mortality generally increased with age for both males/females and urban/rural areas. There was a decrease in the relative risk for males and females aged 0–4 to 5–9 years, and then the mortality increased with advancing age. Females had a relatively higher risk of asthma mortality than males from age 10 to 55, and males had a slightly higher risk after age 60. People in rural areas aged 35 to 75 have a higher risk than those in urban areas, while people in urban areas older than 75 have a higher risk.

For the period effect (Figure 3b), there was a V-shaped trend for both males/females and urban/rural areas from 2004–2008 and 2009–2013 to 2014–2018, with the 2009–2013 period having the lowest risk. The risk in the 2014–2018 period was 1.38, 1.43, 1.22, and 1.56 times the risk in the 2009–2013 period for males, females, urban, and rural areas, respectively.

The cohort effect showed a general decrease in the risk of mortality from earlier to later birth cohorts (Figure 3c), but there was a fluctuation for the cohort born after 2000 for both males/females and urban/rural areas.

## 4. Discussion

From a previous study, asthma was more common in males, which has been explained by differences in lung/airway size and immunology [11,12]. This could also explain why the asthma mortality rate is higher for males than females. Our study showed mortality rate in urban areas was higher than rural areas. This is consistent with an earlier study showing that residents of urban areas are more likely to have asthma than those of rural areas [13,14]. Early exposure to microbes in rural areas might play a role in moderating the immune system to reduce the risk of asthma and allergies. In addition, pollutants, pests, mouse allergens, stress, and the development of obesity in urban environments might cause adverse effects on children’s respiratory health [11,13,14].

The age standardized asthma mortality increase from 2012 to 2013 may be because, in 2013, the National Health and Family Planning Commission took the lead to work with the National Disease Surveillance System to increase the number of monitoring centers from 161 to 605.

This study presented the trend of asthma mortality for all ages by estimating the percent changes in rates and the age, period, and cohort effect from 2004 to 2019. Joinpoint regression analysis showed that the mortality rates decreased in all age groups for both males and females in urban and rural areas. The decreasing trend of the mortality rates has slowed down during 2007 and 2009, after which the decreasing trend lessened. This is similar to a previous study on the trends of international asthma mortality that there was no noticeable change in global asthma mortality rates from 2006 through to 2012 [3]. This may because the widespread use of inhaled corticosteroids (ICS) therapy has reduced the asthma mortality before 2007, and the stabilized trend after 2007 may now be due to the patients with severe asthma contributing to non-preventable deaths. This may suggest to us that we may need to seek for novel asthma management strategies to further reduce the asthma mortality rate in the future. 

Through the APC model with IE methods, the age, period, and cohort effects on the mortality of asthma were discussed here. The age effect on asthma mortality increased with advancing age, which could be mainly contributed by the aging transition of China. Previous studies demonstrated that Chinese aging rose rapidly from 1980 to 2010 [15]. Older people may be under treatment on asthma plans, and asthma may cause other severe diseases for senior people. Therefore, how to implement the asthma management plan or offer new strategies according to the characteristics of senior people to control the asthma mortality rate is significant in reducing the asthma mortality rate for the entire population. On the other hand, the mortality rates for children under 5 years old were relatively higher than those for the other children and adolescents. As stated in previous studies, there is no greater cost than the preventable death of an asthma patient, especially among children [16,17]. Recent studies have demonstrated that asthma in children is increasingly prevalent in China [18]. How we can better implement optimal asthma care management plans and offer intervention delivery to relieve the asthma burden and to reduce asthma mortality, especially for children, is a challenge for our health authorities, practitioners, pharmacists, and other health professions in the future.

The period effect can be influenced by a set of environmental, historical, and economic factors. As the economy and technology developed, there had been a widespread and progressively increasing use of some novel techniques, and other asthma management plans had been shown to reduce the mortality risk in asthma [3,19,20]. The increasing trend of the period from 2009–2013 to 2014–2018 may be partly due to the increase in monitoring centers in 2013. This may also be because urbanization and technological development in recent decades have made the environment worse and unhealthy for people with allergies and asthma.

The cohort effect echoed the influencing factors that were unique to the group, and they went across with time for the specific cohort. The cohort effect on asthma mortality continuously decreased from birth cohort 1925 to birth cohort 2000. This may be because more recent birth cohorts have received better education, have a better awareness of disease prevention, and may have a healthier diet habit than earlier birth cohorts. These factors can generally explain the decreasing cohort effect. However, there was a fluctuation of the cohort trend for the cohort born after 2000. The possible reason was that as industry advances in China, air pollution worsens; people have a higher risk of asthma when exposed to air pollution, especially during the early years of childhood [21].

As the industrialized level of society has developed rapidly in China in recent decades, new technologies have been implemented to manage the prevalence of asthma and reduce the risk of death [22]. However, at the same time, air pollution and some other environmental factors, as well as the aging transition of society, have slowed the decrease in current asthma mortality [23,24].

The limitation of this study was that, although this study investigated the trend and age, period and cohort effects together for the mortality rate due to asthma for the population in China, residual or unmeasured confounders may still have affected results because of the lack of information on various confounders including medical condition. Therefore, there might be some confounding aspects that may affect the mortality trend in the population, especially for the older age of the population. Second, as the monitoring centers increased from 161 to 605 in 2013, there might be some inconsistency in the data collection. Therefore, the results of this study should be treated cautiously. The asthma mortality and related data used in this study were extracted historically from a routine surveillance data [7]. The number of persons who have died from asthma in China may be higher than reported [25]. However, the Chinese mortality surveillance system is the only national mortality surveillance system covering all causes of death in people of all ages. Since 2013, the system has increased the surveillance population from 6% to 24% of the Chinese population, covering almost one quarter of the Chinese population. It is the only practicable option for generating reliable and valid in-formation on asthma mortality in the country.

## 5. Conclusions

To our knowledge, this analysis was the first to study the trend and age, period, and cohort effects together for the mortality rate due to asthma for the population in China. The age, period, and birth cohorts of the population can all contribute to the variance of the asthma mortality rate. This finding can provide some new thoughts and clues to the study of the cause of mortality due to asthma in the current post-industrialized society to suggest novel strategies and optimal asthma management plans to reduce the asthma mortality rate, especially for older people facing higher risk and younger people born during periods with higher levels of industrialization with more environmental problems.

## Figures and Tables

**Figure 1 healthcare-10-00346-f001:**
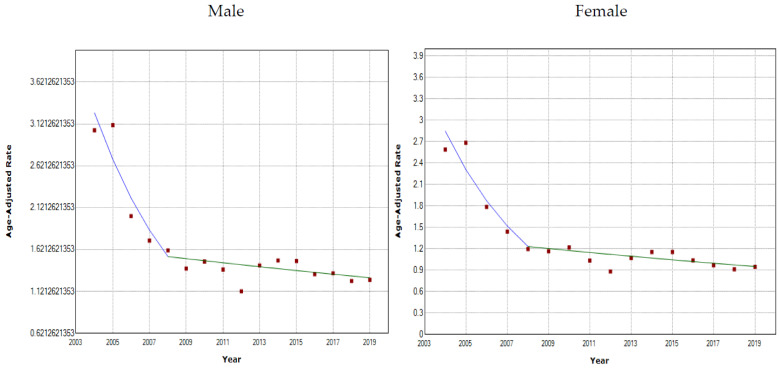
Joinpoint analysis of age-adjusted asthma mortality rate for males and females.

**Figure 2 healthcare-10-00346-f002:**
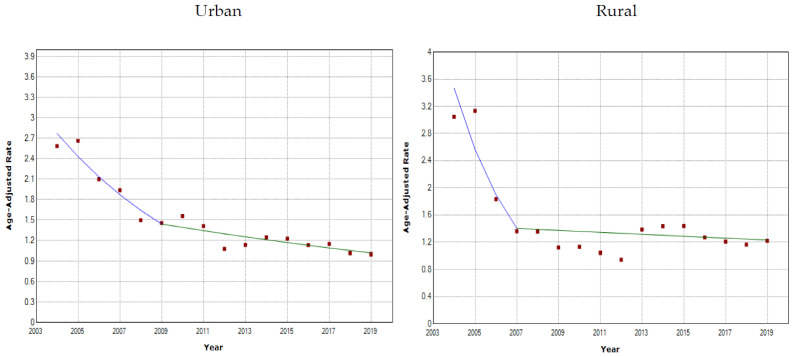
Joinpoint analysis of age-adjusted asthma mortality rate for urban and rural areas.

**Figure 3 healthcare-10-00346-f003:**
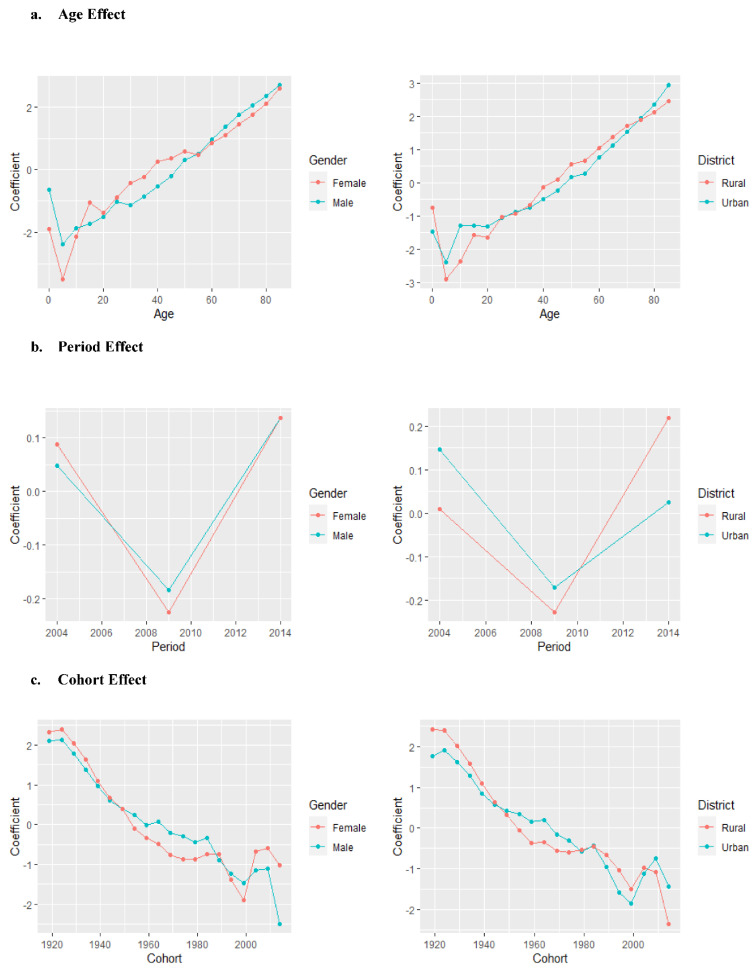
Age-period-cohort effect of asthma mortality from 2004 to 2019 in China by gender/district.

**Table 1 healthcare-10-00346-t001:** Asthma mortality (per 100,000 persons) in China from 2004 to 2019.

Year	ASR	Crude Rate	Men	Women	Urban	Rural	Eastern	Central	Western
2004	2.12	2.42	2.55	2.28	2.60	2.32	2.09	2.92	2.22
2005	2.16	2.49	2.60	2.37	2.74	2.37	2.07	3.24	2.12
2006	1.43	1.69	1.74	1.64	2.28	1.41	1.52	2.27	1.16
2007	1.19	1.45	1.55	1.35	2.18	1.07	1.24	1.87	1.21
2008	1.05	1.33	1.50	1.14	1.81	1.07	1.21	1.52	1.26
2009	0.96	1.15	1.24	1.06	1.72	0.84	1.06	1.18	1.24
2010	1.02	1.20	1.30	1.09	1.74	0.86	1.17	1.09	1.36
2011	0.91	1.15	1.28	1.00	1.47	0.93	1.09	1.03	1.40
2012	0.76	1.12	1.23	1.01	1.37	0.96	1.07	1.05	1.30
2013	0.95	1.52	1.70	1.33	1.61	1.48	1.52	1.36	1.75
2014	1.01	1.66	1.83	1.48	1.73	1.63	1.77	1.47	1.76
2015	1.01	1.66	1.83	1.47	1.72	1.62	1.78	1.26	1.98
2016	0.90	1.62	1.77	1.45	1.65	1.60	1.72	1.27	1.92
2017	0.88	1.59	1.80	1.37	1.73	1.52	1.65	1.29	1.90
2018	0.82	1.59	1.78	1.39	1.66	1.55	1.65	1.37	1.78
2019	0.83	1.58	1.77	1.38	1.67	1.53	1.58	1.48	1.71

ASR: age standardized rate per 100,000 persons (using Chinese standard population in 2000).

**Table 2 healthcare-10-00346-t002:** Annual percent change by gender or district.

Annual Percent Change (APC)								
Cohort	Segment	Lower Endpoint	Upper Endpoint	APC	Lower CI	Upper CI	Test Statistic (t)	Prob > |t|
Male	1	2004	2008	−17.1 *	−25.7	−7.5	−3.8	0.003
Male	2	2008	2019	−1.6	−3.5	0.3	−1.9	0.084
Female	1	2004	2008	−18.9 *	−27.9	−8.8	−3.9	0.002
Female	2	2008	2019	−2.3 *	−4.4	−0.2	−2.4	0.033
Urban	1	2004	2009	−12.3 *	−17.9	−6.2	−4.3	0.001
Urban	2	2009	2019	−3.4 *	−5.2	−1.6	−4.1	0.002
Rural	1	2004	2007	−26.1 *	−42.8	−4.5	−2.6	0.025
Rural	2	2007	2019	−1.1	−3.7	1.6	−0.9	0.392

* Indicates that the annual percent change is significantly different from zero at the alpha = 0.05 level.

## Data Availability

The data presented in this study are available on request from the corresponding author.

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
