# Peer review of "Long-Term Trends of Asthma Mortality in China from 2000 to 2019: A Joinpoint Regression and Age-Period-Cohort Analysis"

_healthcare, 2022, doi:10.3390/healthcare10020346_

Round 1

Reviewer 1 Report

This manuscript provides us information about the trends of asthma mortality in China.   Although local, this study give us a little perspective about the reality in China and maybe can help to improve the asthma management plans to reduce the asthma mortality rate. The article has a clear language and the aim of the study it is clear and interesting. However, there are some problems, which must be solved to improve the manuscript quality.

I would like to make a few suggestions for revision:

Abstract

Line 11- ”The aim of this study was to assess the trends of asthma mortality in China. (2)”. Normally we shouldn't put references in abstracts. Please revise this point.

Line 15- “ mor-tality”. The correct it is mortality, all the letters together. The authors have several words, in all the text, with the same mistake. Please, revise all the text.

Line 18– “An important change took place between 2007 and 2009”- The authors need to clarify these phrase, because, as it is written, it does not give any information

Materials and methods

Line 47-“ The asthma mortality and related data for this study were extracted from National Death Cause Datasets of Disease Surveillance System written by Chinese Center for Disease Control and Prevention, which is considered to be representative nationally as it is a population-based death registration system containing 300 million people across 31 provinces using an iterative method involving multistage stratification[7].” It's a very long sentence, please rephrase this sentence

Results

Line 126-“ ). From a previous study, asthma was more common in males, which has been explained by differences in lung/airway size and im-munology[11, 12]. This could also explain why the asthma mortality rate is higher for males than females.”

Line 131 –“ This is consistent with an earlier study showing that residents of urban areas are more likely to have asthma than those of rural areas[13, 14]”

In my point of view these two paragraphs should be moved to the discussion, because it is a possible explanation for the results.

Table 1-Authors also must standardize the decimal places used- this is because authors mostly use 2.28 (for example) with two decimal places, but then they also use 1, or 2.6...

Reviewer 2 Report

Thanks to the author(s) for this contribution examining Long-term Trends of asthma mortality in China. The manuscript indicates that asthma mortality rates generally have a positive relationship with the age of the population when controlling for period and cohort but a supportive asthma management plan is need for under age 5 and older adults.

Comments:

Result: Line 165, Figure 1.3A should be referred.

Discussion:

  1. Line 190: “The increase from 2012 to 2013 may be because in 2013”, what increase? Adding “The rate increase” will make it clear
  2. Line 207-209: “As stated in previous 207 studies, it is a great cost if the death of a patient can be prevented, especially among chil-208 dren [16, 17], and it is not impossible to reduce asthma mortality.”  This sentence was cleared stated, revision need.
  3. paragraph 2 just re-stated the result from the joint point regression, expand the discussion for joint point regression results could help readers better understand the impacts of these analysis results. For examples: what are the possible causes for deceasing mortality rates, why male and female, urban and rural showed the similar mortality rate trend pattern, comparing with reported mortality rate trend from other counties, whether the trend showed the similar pattern?

Conclusion:

Three steps have been used for analyzing the data; however, the conclusion only summarized the result from Age-period-cohort (APC) analysis with the intrinsic estimator method.  Results from Description analysis and Joinpoint regression analysis didn’t cover into the conclusion.

Author Response

Please see the attachment. Thanks! 

This manuscript is a resubmission of an earlier submission. The following is a list of the peer review reports and author responses from that submission.